# BEYOND PRUNING: NEURO-INSPIRED SPARSE TRAINING FOR ENHANCING MODEL PERFORMANCE, CONVERGENCE SPEED, AND TRAINING STABILITY

## ABSTRACT

Pruning often trades accuracy for efficiency, and sparse training is hard to do from scratch without performance loss. We introduce a simple, neuro-inspired sparse training (NIST) algorithm that simultaneously sparsifies, stabilizes, and improves neural networks without introducing computational overhead or complex optimizers. Our method achieves high sparsity while surprisingly enhancing model performance, accelerating convergence, and improving training stability across diverse architectures, and plugs directly into standard training pipelines. Empirically, it strengthens MLP-heavy architectures (e.g., VGG, AlexNet) by aggressively sparsifying them (>90%) and at the same time, improving test accuracy by 8-10%. Additionally, NIST accelerates convergence and reduces variance in efficient CNNs such as MobileNet. It also enables transformer training directly from 50% initial sparsity and up to 70% final sparsity with negligible performance loss, while speeding up model convergence in the first 30 epochs. Our comprehensive experiments, ablations, and comparisons against state-of-the-art pruning and sparse-training methods reveal that these gains stem not from reduced parameter counts alone, but from improved optimization dynamics and more effective parameter reallocation. This study reframes sparse training as a performance-enhancing tool rather than a compromise.

## 1 INTRODUCTION

Biological nervous systems operate with sparse, highly structured connectivity that yields efficient, robust computation under tight resource constraints Hole & Ahmad (2021); Sprenger (2008); Petanjek et al. (2023); Stiles & Jernigan (2010). By contrast, state-of-the-art artificial neural networks rely on dense, over-parameterized layers that drive substantial compute and memory costs. In this work, we ask whether the efficiency of brain-inspired sparsity can be reconciled with the expressivity of modern deep models: can simple, architecture-level sparse ANN connectivity deliver large reductions in computational cost while preserving—or even improving—learning performance?

During early brain development, neurons form many provisional branches and synapses, and experience-driven mechanisms then remove redundant connections to stabilize function and improve efficiency (Benson, 2020; Gyllenhammer et al., 2022; Holt & Mikati, 2011; Kostović et al., 2019; Lohmann & Kessels, 2014; Tierney & Nelson, 2009). Such motif-proliferation followed by selective refinement—inspires our approach of structurally initializing wide connectivity and then applying a targeted, single-shot sparsification to obtain compact, stable models.

Large models and dense layers are a primary source of the compute and memory burden in deep learning Li et al. (2023); Cheng et al. (2024); Liebenwein et al. (2021). A variety of sparsification techniques have been developed to reduce the computational burden of neural network training Cheng et al. (2024); Hoefler et al. (2021). Prune-at-init (PaI) methods identify compact subnetworks using saliency or gradient signals before training, while dynamic sparse training (DST) alternates pruning and regrowth during optimization Nowak et al. (2024); Hoefler et al. (2021); Jiao et al. (2022). Other dynamic sparsification or pruning approaches often rely on heuristic mask updates or require additional passes/optimizer modifications to tune mask and weight separately Ji et al. (2024).

Although effective, these methods often introduce complexity through additional computations, topology updates, or hyperparameters, making large-scale deployment challenging Jiao et al. (2022); Hoefler et al. (2021). This has led to interest in sparse-from-scratch methods that maintain the benefits of sparsity, efficiency, regularization, and robustness, while being simple, reproducible, and easily integrated into standard training pipelines with minimal overhead.

We propose Neurogenesis-Inspired Sparse Training (NIST), a structurally simple and zero-cost sparse training pipeline for fully connected layers. Each layer receives a small integer hyperparameter, the *neuroseed factor*, which deterministically specifies the number of outgoing synapses per input feature. We then apply a fixed, locally structured binary mask at initialization (e.g., half the weights pruned), train on the masked network, and — optionally — perform a single-shot magnitude-based pruning step mid-training to further sparsify. The updated mask is then kept fixed for the remainder of training. Crucially, NIST requires no saliency scoring, no iterative regrowth, and no specialized optimizer state; it adds no measurable computational overhead beyond masked forward/backward arithmetic.

The method developed here is grounded in the topographic sparse mapping framework proposed by Kamelian Rad et al. (2025), where sparsity was introduced only at the input layer. In contrast, the present work **extends this structural principle to *all* layers**, producing a fully deterministic, data-free sparse connectivity pattern. This generalization transforms the earlier single-layer mechanism into a unified pruning strategy that governs the entire network architecture.

We briefly summarize the distinctions of our approach and defer full comparison to Section 2 and Section A.1. In short, NIST is a zero-cost, structural, sparse training recipe: it uses deterministic bio-inspired masks at initialization, requires at most one magnitude-based pruning step, and thereafter trains with a fixed sparse topology using standard optimizers. Empirically, this simple pipeline not only reduces parameter and floating point operations (FLOPs) counts substantially, but also yields improved convergence speed and reduced run-to-run variability in many settings; complete comparisons and ablations are given in Section A.4 and Section 4.

The scope of this study goes beyond proposing another sparse-training recipe. Through systematic experiments on CNNs and transformers (e.g., LeNet-5, VGG, AlexNet, MobileNet, DeiT_Small), we show that simple, zero-cost structural initializations combined with a single-shot magnitude prune can accelerate convergence, reduce run-to-run variability, and in some cases improve final accuracy relative to dense baselines. Equally important, we introduce a novel design maneuver—replacing a single dense head with multiple sparse heads—to reallocate a fixed parameter budget more effectively; we show that this architectural reallocation consistently improves stability and generalization without adding computational overhead. In short, our contribution is both empirical and conceptual: a practical, easy-to-adopt sparsification recipe plus evidence that sparsity can act as an inductive bias and design primitive, not merely as a savings mechanism.

Notably, on DeiT_Small trained from scratch on ImageNet-100, initializing at 50% sparsity and applying a one-shot prune to 60% yields a drop of $< 0.3\%$ top-1 accuracy compared to the dense model while reducing parameter count and FLOPs substantially. On smaller CNN benchmarks (e.g., LeNet-5 on CIFAR-10), aggressive sparsification (up to 99.5%) preserves or even improves validation accuracy and noticeably reduces overfitting. Across VGG and AlexNet experiments, we demonstrate several orders-of-magnitude reductions in active weights and FLOPs with substantial accuracy gains. Finally, we show that sparse architectural reallocation (e.g., replacing a dense head with multi-layer sparse heads) can further improve stability and accuracy under tight parameter budgets.

In summary, NIST provides a practical, zero-overhead recipe for large-scale sparsification: simple deterministic masks, one-shot pruning, and masked training. The method is easy to implement, reproducible, and broadly compatible with standard optimizers and training pipelines. The remainder of the paper presents the method in detail (Section 3) and experimental evaluation (Section 4).

## 2 COMPARISON TO PRIOR WORK AND CONTRIBUTIONS .

In contrast to prior sparsification approaches, which often rely on complex optimization procedures, iterative pruning–regrowth cycles, or gradient-based mask updates, our method achieves state-of-

the-art compression with no added computational overhead. This behavior and the design choices distinguish our approach along four axes:

- **Zero-Cost Structural Initialization.** Our method departs from prune-at-init (PaI) approaches by abandoning any per-weight saliency scoring or extra initialization passes Frankle et al. (2020); Wang et al. (2021). PaI methods (e.g., SNIP Lee et al. (2019), GraSP Wang et al. (2020); Zhang et al. (2022)) evaluate data- or gradient-dependent importance to select connections prior to training, which requires additional forward/backward computations and per-weight scores at initialization Liu et al. (2024b); Hoefler et al. (2021); Jaiswal et al. (2022). In contrast, we use a deterministic, structural mask (half the weights per layer in the typical configuration) that is applied once at the start. This design requires no extra passes, no score computations, and yields perfectly reproducible connectivity, i.e., the mask is a low-cost architectural choice rather than an expensive selection procedure.

- **Single-shot pruning (no dynamic growth) vs. dynamic sparse training (DST).** Our pipeline uses at most one magnitude-based pruning step and then fixes the binary mask for the remainder of training, which contrasts sharply with dynamic sparse training (DST) methods that alternate pruning and regrowth (e.g., RigL Evci et al. (2020) and related algorithms) Hoefler et al. (2021); Nowak et al. (2024); Vadera & Ameen (2022). DST requires continual score evaluation, topology updates, and bookkeeping (and often specialized hyperparameters and update rules), all of which introduce runtime and implementation overhead Wang et al. (2024); Hoefler et al. (2021); Lasby et al. (2023). By eliminating regrowth and any per-step topology optimization, our approach preserves the simplicity of standard training loops: it works with off-the-shelf optimizers and masked arithmetic only, and thus adds no measurable computational overhead beyond the masked forward/backward passes.

- **Breaking the Performance–Sparsity Trade-off.** Notably, our method delivers counterintuitive empirical benefits that go beyond efficient compression. Structural initialization plus single-shot pruning (i) revives over-parameterized networks by improving convergence speed and, in many cases, boosting final accuracy relative to the dense baseline, (ii) increases training stability by reducing run-to-run variability, and (iii) in some architectures breaks the conventional accuracy–sparsity trade-off: aggressive pruning can yield equal or better accuracy than the dense model. These effects indicate that the masks are not merely a parameter-reduction technique but introduce a strong inductive bias that regularizes learning and improves optimization dynamics, all without extra computations, optimizer changes, or complex topology heuristics.

- **Sparsity as Regularization and Parameter Reallocation.** Beyond compression, our approach demonstrates that structural sparsity can act as an effective regularizer and enable smarter parameter allocation. For instance, replacing a dense classifier head with multi-layered sparse heads preserves or even improves accuracy while using the same or fewer parameters. This architectural tweak reduces overfitting, improves run-to-run training stability, and guides the network to allocate representational capacity more efficiently. In essence, sparsity is not just a mechanism for pruning weights—it reshapes the network's functional structure to enhance optimization dynamics and model generalization without additional computational cost.

- **Comparisons and Ablations.** Comparisons with data-driven pruning (e.g., SNIP, SET, RigL) and the ablation studies have **already been thoroughly addressed in a comprehensive companion study** Kamelian Rad et al. (2025). That work includes:
    - **explicit head-to-head evaluations** against SNIP, SET, RigL, CTRE, and other state-of-the-art pruning methods,
    - **full ablations** showing why the topographic choice of $k = 1$ in the first layer is the most effective configuration,
    - **comparisons with random masks** and parameter-matched dense baselines.

  These findings collectively demonstrate that deterministic structural sparsity is **competitive with, and often superior to, data-driven pruning** at matched sparsities.

NIST sparsely initializes all layers using a neuro-inspired connectivity and mimics the neurodevelopmental dynamics of the mammalian brain. This framework is novel to the best of our knowledge in the field of artificial neural networks. While there are studies exploring biologically inspired

sparse training and dynamic connectivity, none have implemented a fixed, uniform neuroseed factor per layer as we propose.

## 3 METHOD

### 3.1 NEUROSEED INITIALIZATION AND CONNECTIVITY MASK

We introduce a layer-wise integer hyperparameter, the *neuroseed factor* $k \in \{1, \ldots, N_{\text{out}}\}$, which controls the number of outgoing synapses per input feature for a linear layer with $N_{\text{in}}$ inputs and $N_{\text{out}}$ outputs. Connectivity is deterministic and homogeneous: each input index $i \in \{0, \ldots, N_{\text{in}}-1\}$ connects to exactly $k$ output indices using a wrap-around (modulo) rule. The initialized connectivity between input and output units can be represented as a binary mask. Formally, the binary mask $M \in \{0, 1\}^{N_{\text{in}} \times N_{\text{out}}}$ is defined as Eq. 1:

$$M_{i,j} = \begin{cases} 1, & \text{if } j \in \{(i+t) \bmod N_{\text{out}} \mid t = 0, \ldots, k-1\}, \\ 0, & \text{otherwise.} \end{cases} \tag{1}$$

The layer weight matrix $W$, initialized by Kaiming uniform distribution, is masked element-wise at the start as Eq. 2:

$$\widetilde{W} = W \odot M, \tag{2}$$

and training proceeds directly on $\widetilde{W}$. This deterministic top–down / contiguous routing preserves signal propagation and avoids narrow bottlenecks while providing strong sparsity and reproducibility guarantees.

**Implementation (mask generation).** We implement the mask as follows (indices: inputs → rows, outputs → columns): self.indim and self.outdim correspond to the input and output dimensions, respectively.

```
for i in range(self.indim):
    for t in range(self.neuroseed_factor):
        j = (i + t) % self.outdim
        mask[i, j] = 1
```

**Special cases and topographical maps.** Setting $k = 1$ gives each input feature a single deterministic target output (minimal nonzero connectivity), which minimizes density while avoiding feature loss. This case is inspired by retinal (biological) topography (see Appendix A.2 for more information). Larger $k$ increases per-feature redundancy and robustness.

**Pruning schedule and training protocol.** We optionally apply a single-shot, permanent magnitude-based pruning stage after the model has partially stabilized (empirically when it reaches $\sim 50\%$ of its eventual peak accuracy). As a practical rule we prune after $\sim 10\%$ of the total training epochs. At the pruning step we zero the smallest-magnitude weights, update $M$ accordingly, and keep the updated mask fixed for the remainder of training. This pipeline:

$$\text{mask init} \rightarrow \text{(optional) single-shot pruning} \rightarrow \text{fixed-mask training}$$

does not require regrowth, does not maintain dense connectivity representations, and adds no algorithmic overhead beyond masked arithmetic. Example schedules used in our experiments:

- VGG16 (frozen convolutional base): prune after 4 epochs.
- DeiT trained from scratch for 300 epochs: prune between epochs 50–80.

**Practical recommendations.**

- CNNs: set $k = 1$ for the first (input) layer, then use half density in deeper layers, a heuristic from ablation experiments we found sufficient to preserve dense-level complexity without hurting initial performance Kamelian Rad et al. (2025).

- Transformers: initialize all linear layers (e.g., QKV, projection, FC, etc) at approximately half density (choose $k$ so each input connects to $\sim 50\%$ of outputs), but keep the classifier head dense.

- Stacked MLPs: use $k = 1$ for the first linear layer (mimicking retinotopy).

These heuristics are motivated by topographical mapping and biological neurodevelopment (initial overproduction followed by activity-dependent pruning) and were found to maximize sparsity without degrading performance in our experiments. See Figure 1 for an illustration of the mapping Kamelian Rad et al. (2025).

With this framework, we first initialize the input layer with topographical mapping. The remaining layers are populated to a sufficient density before training begins, as dictated by the selected neuroseed factor. One effective strategy is to use a half-dense configuration, where the input layer is fully dense, and subsequent layers are initialized with 50% connectivity. Training then begins, and pruning is applied after a few epochs. This approach enables the network to achieve a high degree of sparsity from the very beginning, improving computational efficiency. Furthermore, this strategy mirrors aspects of biological neurodevelopment: networks start with dense connectivity and gradually refine their structure by eliminating redundant connections during learning.

Figure 1 illustrates two layers with 10 and 5 neurons. Each neuron in the first layer can potentially form up to 5 connections, yielding all-to-all connectivity. Sparsity is introduced via the neuroseed factor, a hyperparameter controlling the number of connections per neuron. Connectivity begins as a one-to-one top–down mapping, with synapses growing uniformly in the same direction, expanding systematically across layers.

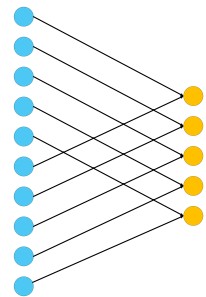 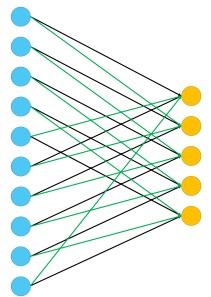 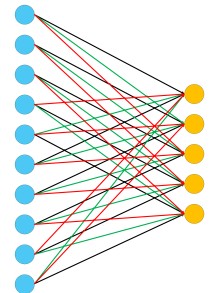

(A) Neuroseed Factor=1      (B) Neuroseed Factor=2      (C) Neuroseed Factor=3

Figure 1: Illustration of the initial connectivity pattern determined by the neuroseed factor k across two layers comprising 10 and 5 neurons, respectively. A) With a neuroseed factor of k=1, each presynaptic neuron establishes a single synapse, forming a topographic mapping that is particularly suited for input mapping when we have stacked MLP layers. B) Increasing the neuroseed factor to k=2 allows each presynaptic neuron to form an additional connection with the neuron directly below the previous target in the postsynaptic layer, promoting localized but expanded connectivity. C) A neuroseed factor of 3 extends this homogeneous growth, enabling each presynaptic neuron to connect with three consecutive neurons in the next layer, further increasing initial synaptic coverage in a structured manner.

## 4 EXPERIMENTS

We evaluated NIST across various vision benchmarks to test its ability to sparsify fully connected classifier heads without compromising performance. Our experiments are designed to answer two questions: (i) What does NIST provide beyond just "sparsification"? and (ii) Are these effects consistent across different architectures and data scales? We report both final metrics (accuracy, compression ratio) and training dynamics (learning curves, stability, training cost), with further ablations and extended results provided in the Appendix A.4.

Training pipeline with NIST

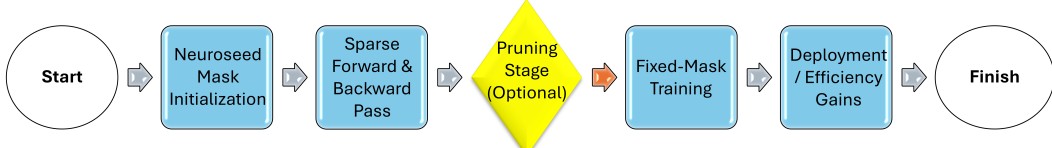

Figure 2: Overview of the proposed framework. 1) Generate a deterministic binary mask using the neuroseed factor; apply it to weight matrices before training. 2) Train directly on masked weights without regrowth or dense overhead. 3) After $\sim 10\%$ of training epochs (e.g., 4 in VGG16), apply one-shot permanent magnitude pruning to refine connectivity. 4) Continue training with the updated sparse mask; no further changes in connectivity. 5) Final sparse model achieves reduced FLOPs and memory footprint while retaining/improving accuracy.

## 4.1 SETUP

**Datasets.** Experiments were performed on FMNIST, CIFAR-10, and ImageNet-100. We employed dataset-specific preprocessing strategies, ranging from minimal to heavy augmentation.

**Architectures.** We evaluate NIST on LeNet-5, AlexNet, VGG16, MobileNet, and DeiT-Small.

**Training details.** Optimizer, schedules, batch sizes, and compute are detailed in Appendix A.3. For ImageNet-100 on DeiT, we report 5-run results for each case due to compute constraints; variance between runs is small.

**Evaluation metrics.** We report Top-1 accuracy, classifier compression ratios, epochs-to-converge, training FLOPS, and for MobileNet, we compute stability metrics (% epochs with lower SD and AUC_SD).

## 4.2 MAIN RESULTS

Table 1 summarizes performance, compression, and computational cost (FLOPs). NIST compresses classifier heads of AlexNet and VGG16 by 10,000 and 1,000 times, respectively, while improving Top-1 accuracy. Notably, VGG16 improves from $71.2\% \rightarrow 78.2\%$ with $>99\%$ classifier sparsity and $\sim 89.5\%$ overall (see Appendix A.4 for detailed analysis). For DeiT-Small, training from 50% sparsity achieves near-dense accuracy with negligible loss even under further pruning.

In Fig. 3A, the validation accuracy trajectories of the original dense VGG16 with its NIST-sparsified counterparts are compared under different training configurations. The dense baseline reaches a plateau near 71% validation accuracy. The dense model converges slowly with a low learning rate, while a higher rate accelerates convergence but induces oscillations, likely due to fine-tuning the head on a frozen base.

Similarly, Fig. 3B illustrates the mean validation accuracy across 400 training epochs for DeiT trained with and without NIST. While the DeiT baseline (red) benefits from dense training, NIST-equipped models consistently close the gap, achieving comparable or even superior performance at initial epochs (faster convergence, see Appendix A.6). For models trained toward extreme sparsity (95% final density), when the pruning stage is performed at epoch 200, the accuracy initially drops due to significant parameter loss, but then rises rapidly, demonstrating the network's capacity to recover and adapt under aggressive sparsification. These results highlight the robustness of NIST, showing that even with severe parameter reductions, the network maintains competitive accuracy.

**Reviving Over-Parameterized Architectures (VGG16, AlexNet).** All NIST-sparsified configurations of VGG16 and AlexNet exhibit both faster convergence and stronger generalization. This improvement highlights two critical insights. First, sparsification under NIST not only preserves accuracy but actively improves generalization, challenging the assumption that pruning or sparsity necessarily entails performance loss. Second, the parameter reduction is dramatic: VGG16 is reduced from $\sim 134$million parameters to $\sim 14$ million, while boosting ImageNet-100 top-1 accuracy

Table 1: Summary of model compression and final Top-1 accuracy across representative models. Compression = dense_params / nist_params. "Epochs to converge" is epochs until validation Top-1 reaches 95% of final value.

| Model | Dataset | Params | Compression | FLOPs($\times 10^{12}$) | Top-1 (%) | Epochs |
|---|---|---|---|---|---|---|
| AlexNet (Dense) | CIFAR-10 | 33.6M(Head) 35.8M(Total) | 1$\times$ | 262 | 79.4 | 15 |
| AlexNet + NIST | CIFAR-10 | 3.3k(Head) 2.25M(Total) | 10,000$\times$(Head) 15.8$\times$(Total) | 0.012 | 86.4 | 7 |
| VGG16 (Dense) | ImageNet-100 | 120M(Head) 134M(Total) | 1$\times$ | 959.9 | 71.4 | 20 |
| VGG16 + NIST | ImageNet-100 | 120k(Head) 14M(Total) | 1,000$\times$(Head) 9.5$\times$(Total) | 0.31 | 78.6 | 5 |
| DeiT_Small (Dense) | ImageNet-100 | 22M | 1$\times$ | 4,576 | 81.91 | 400 |
| DeiT_Small + NIST | ImageNet-100 | 8.8M | 2.5$\times$ | 2,059 | 81.68 | 400 |
| DeiT_Small + NIST | ImageNet-100 | 6.6 | 3.3$\times$ | 686 | 81.38 | 400 |

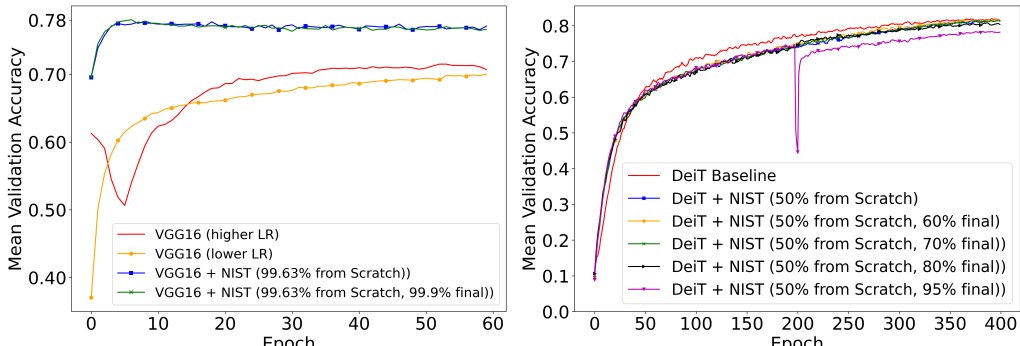

(A) Learning Curves: VGG16 Baseline vs NIST-Sparse Head Variant

(B) Learning Curves: DeiT_Small Baseline vs NIST variant

Figure 3: Performance comparison of training with NIST. A) Validation accuracy of VGG16 with and without NIST. Pruning 99.9% of VGG16's ≈120M-parameter head boosts validation accuracy from 71.4% to 78.5%. Moreover, VGG trained with NIST converges far faster and more reliably, reaching 70% accuracy in a single epoch and its peak accuracy within 5 epochs, while the dense model struggles to converge under its heavy parameter load. B) Validation accuracy of DeiT_small with and without NIST. Starting from 50% sparse initialization, extending to different target densities (60%, 70%, 80%, 95%). At 95% final sparsity, accuracy briefly dips at epoch 200 when pruning is applied, but quickly rebounds. Across sparsity levels, NIST matches the dense baseline despite drastically reduced parameter budgets.

by 7 percentage points. In effect, NIST breathes new life into VGG16—an architecture widely considered outdated—by making it both drastically lighter and substantially more accurate, rivaling or even surpassing modern efficiency-focused ANNs.

**Seamless Sparsification of Modern Transformers (DeiT-Small).** Table 1 and Fig. 3B report results for DeiT_Small trained on ImageNet-100 for 400 epochs under six sparsification regimes. The dense baseline achieves 81.90% top-1 accuracy. Initializing the network at 50% sparsity from scratch (NIST Half Sparse) yields nearly identical performance (81.48%), confirming that deterministic structural masking does not harm convergence. Applying NIST's single-shot pruning stage at epoch 200 further reduces the active weights while maintaining accuracy: at 60% sparsity, accuracy remains 81.69%, and even at 70% sparsity it holds at 81.39%, within 0.5% of dense. Performance begins to decline more noticeably at higher sparsity levels, with 80% yielding 80.81% and 95% dropping to 78.37%.

Importantly, these gains are achieved without iterative regrowth or specialized optimization: only structural initialization and one pruning step. In terms of efficiency, NIST compresses DeiT_Small

from 22M to 8.8M parameters (2.5×) and cuts FLOPs by more than half (4,576×10$^{12}$ to 2,059×10$^{12}$) while preserving accuracy, with further reductions possible at higher sparsities. These results demonstrate that modern transformers can be sparsified seamlessly and trained stably at scale with negligible accuracy loss.

### 4.3 BEYOND SPARSIFICATION: MULTI-LAYER SPARSE HEADS.

Here, we investigate whether expanding the classifier head and applying NIST-based sparsification can improve learning dynamics without increasing parameter count. In this setup, the expanded sparse classifier head is trained sparse-from-scratch while maintaining the same, or fewer, parameters compared to the single-layer dense head. Specifically, we replace MobileNetV1's standard single-layer dense head with a three-layer sparse head trained sparse-from-scratch (see Appendix A.5 for detailed analysis).

**Stability Gains from Multi-Layer Sparse Heads (MobileNet).** Fig. 4 compares validation accuracy across configurations: the dense baseline (red), a single NIST-sparse layer (black), and the three-layer NIST head (blue). The Sparse-3 model, corresponding to a classifier head made by 3 sparse layers, converges faster, generalizes better, and exhibits lower variability than the dense head. Quantitatively, Sparse-3 reduced mean epochwise training SD by 37.3% (0.0163 vs 0.0259), with significantly lower variance in 80% of epochs ($p = 4.76 \times 10^{-4}$, Wilcoxon). These results demonstrate that multi-layer NIST-sparse heads not only maintain parameter efficiency but also yield more stable and consistent training.

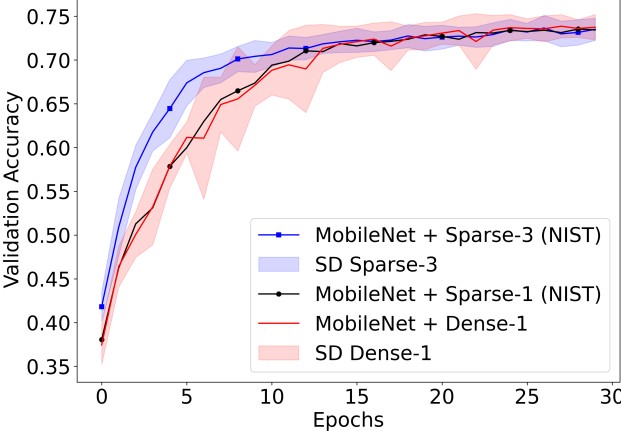

Figure 4: Validation accuracy vs. epoch (mean ± SD over 20 runs) for MobileNetV1 with different classifier heads: Dense-1 (red), Sparse-1 NIST (black), and Sparse-3 NIST (blue). Sparse-3 uses three layers (sizes 100, 200, 10; factors 2, 50, 10) totaling 9,048 params vs. 10,240 for Dense-1. Sparse-3 converges faster and shows lower variance.

## 5 DISCUSSION

Our findings suggest that sparsification can move beyond efficiency tricks: when applied in a biologically inspired way, it can fundamentally alter model behavior, improves over-parameterized CNNs, and simplify modern transformers without added cost. Acting as an implicit regularization tool with no computational overhead, NIST sharpens generalization, convergence speed, and performance while reducing training and inference computational and energy costs.

**Deterministic vs. Data-Driven Sparsity.** Contrary to the assumption that determinism may limit performance, the results of Kamelian Rad et al. (2025) show that deterministically constructed topographic masks can match or exceed the accuracy of data-driven pruning methods such as SNIP. This indicates that structured sparsity does not restrict the discovery of high-quality subnetworks; instead, it offers a low-variance, computation-free alternative to saliency-based pruning.

**Performance gain with zero optimization overhead.**    NIST is a one-shot or light-touch rule: no extra retraining loops, no specialized loss terms, and no second-order estimates. This simplicity reduces implementation friction and makes the method practical for resource-constrained settings. NIST-sparsified models converge faster and outperform dense baselines in overparameterized CNNs, despite using far fewer parameters. On AlexNet, sparsifying the classifier head by 99.99%, the highest sparse regime, exhibited the highest accuracy (79.4% to 86.4%). Similarly, pruning 99.9% of VGG16's $\sim 120M$-parameter head improved validation accuracy from 71.4% to 78.5%, using only 0.03% of the FLOPS of the dense (see Appendix A.4.4).

**Reviving and improving simple CNNs and complementing modern architectures.**    VGG16's straightforward convolutional stacks are fast to train and easy to interpret; combined with NIST they become highly competitive again. This is important: in low-data or low-compute regimes, sparsified CNNs can match or approach transformer-level accuracy without the long training schedules and heavy augmentations that ViTs require. Conversely, for efficient modern models (MobileNet, ResNet), the main contribution is not raw parameter cutting but architectural re-thinking of the classifier head.

**Multi-layer Sparse Heads Stabilize Training.**    Replacing a single dense classifier with a multi-layer sparse head (same or smaller parameter budget) adds representational depth without meaningful cost. For lightweight models this yields faster convergence, lower run-to-run variance, and equal-or-better final accuracy (see Appendix A.5).

**Practical benefits & deployment implications.**    With fewer parameters, models have smaller file sizes and lower memory footprints, and on hardware supporting sparse arithmetic, inference and energy consumption are significantly reduced. Sparsified VGG variants or NIST-headed lightweight models can run efficiently on edge and mobile devices without requiring custom accelerators. The high degree of sparsity also improves interpretability, as visualizing the remaining filters and activations becomes easier for analysis and debugging. Moreover, NIST's simplicity ensures compatibility with other compression techniques such as quantization and distillation.

**Transformers can be sparsified from scratch with minimal cost.**    Our CNN experiments show NIST improves both convergence speed and accuracy, while results on transformers demonstrate its broad applicability. On DeiT-Small, NIST delivers 2.5–3.3× parameter compression with accuracy comparable to dense baselines, without architectural changes. Despite transformers being heavily optimized for dense training, NIST maintains performance, suggesting it is a general sparse training paradigm that compresses and accelerates transformer-based architectures as well.

**Efficient alternative to data-hungry transformers with NIST.**    With NIST, VGG16 compresses from 134M to 14M parameters while accuracy improves from 71.4% to 78.6%. Remarkably, this rivals DeiT-Small's 81.9% despite using fewer parameters, far less training, and minimal augmentation. Thus, NIST can make classical CNNs competitive with transformers, enabling efficient, high-accuracy deployment on modest hardware or useful for data-scarce scenarios.

**Limitations.**    Currently, our recommendations for neuroseed factors are heuristic; automating and optimizing these hyperparameters per layer would likely yield improved and more consistent gains. Additionally, practical runtime improvements depend on hardware and sparse-matrix support—parameter reductions alone do not guarantee speedups across all platforms.

**Directions for future work.**    Promising next steps include: (i) Exploring layer-wise neuroseed configurations (e.g., sparser or denser QKV and feedforward modules) to maximize initial sparsity, (ii) combining NIST with complementary compression methods (quantization, distillation, token sparsification) to maximize end-to-end gains, and (iii) integrating NIST with hardware-aware sparse kernels to realize energy gains.

## 6    CONCLUSION

Sparse training, as demonstrated through NIST, offers far more than a simple reduction in parameter counts. Our results show that models can be aggressively pruned while improving performance,

training speed, and run-to-run variability. We showed that sparsification can act as a constructive force, revealing optimal subnets, considerably reducing computational costs, and even reviving over-parameterized architectures. This reframes sparsity not as a compromise, but as a tool for improved performance, faster convergence, and broader deployment.

Our findings suggest that embracing neuro-inspired sparse connectivity can make both classic CNNs and modern transformers more efficient, interpretable, and practical for real-world use. Looking ahead, automating the choice of neuroseed factors and expanding hardware-level support for sparse computation will further unlock these benefits. Also, our findings suggest that revisiting older, simpler architectures under the lens of sparsity may provide viable and sustainable alternatives to data-hungry, resource-intensive models. We hope this sparks further exploration into neuro-inspired strategies that make deep learning both more powerful and more accessible

ACKNOWLEDGMENTS

**LLM Use.** LLMs have been used solely for the grammatical and spelling checks of the text of this manuscript.

**Reproducibility Statement.** We implemented NIST in both TensorFlow and PyTorch using standard training pipelines to ensure accessibility and reproducibility. All experiments use publicly available datasets (CIFAR-10, ImageNet-100) and common architectures (AlexNet, VGG16, DeiT-Small), with code, hyperparameters, and training scripts provided to replicate results. No custom operators or uncommon libraries are required, making it straightforward to reproduce.

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

# A   APPENDIX

This appendix provides additional details to complement the main paper. First, we expand on the methodological aspects that were only briefly discussed in the main text. Then, we summarize the training details, including optimizers, learning rate schedules, and other implementation specifics, to facilitate reproducibility, followed by extended and detailed experimental results and analyses. Finally, we summarize the ...., including , to .....

## A.1   RELATED WORK

In general, dynamic sparse training (DST) methods, such as RigL Evci et al. (2020), dynamically adjust the sparse connectivity of neural networks during training. These approaches focus on evolving the network's structure over time rather than initializing with a fixed neuroseed factor. They also incorporate a regrowth phase during training which further adds to the complexity and overhead Cheng et al. (2024); Hoefler et al. (2021). Prior work has demonstrated that most DST methods exhibit comparable performance, with weight magnitude emerging as the most effective criterion for pruning Nowak et al. (2024); Zheng et al. (2024); Finlinson & Moschoyiannis (2021); Tmamna et al. (2024).

Cosine Similarity-Based and Random Topology Exploration (CTRE) evolves sparse neural networks by adding connections based on cosine similarity between neurons, inspired by Hebbian learning. However, it does not employ a fixed neuroseed factor per layer. It also imposes additional calculations and computational overhead to calculate the similarities between neurons Atashgahi et al. (2022).

NeuroFabric is another framework that investigates various sparse topologies for neural networks, aiming to identify ideal structures for training. It explores different sparse configurations, while NIST is a one-shot pruning without any extra costs Isakov & Kinsy (2020).

Sensitivity-Based Pruning (SBP), proposed by Hayou et al., estimates the criticality of individual weights at initialization by quantifying their contribution to the loss landscape Hayou et al. (2020). This framework mitigates difficulties encountered when pruning at very early stages, such as the risk of removing entire functional layers, which would compromise network trainability.

Tang et al. introduced Automatic Sparse Connectivity Learning, which restructures network connectivity during training by reparameterizing weights and optimizing sparse patterns Tang et al. (2022). While this method adapts connectivity for performance, NIST pursues a different objective: maintaining biological plausibility and energy efficiency by training under high sparsity from the outset and suddenly advancing to extreme sparsity levels.

A large body of sparsification research begins with networks initialized from a randomly connected topology (random seed). The connectivity is then refined either through random weight additions or by applying selection rules based on diverse metrics, such as gradient signals, weight magnitude, correlations, neuron–weight similarity, or sensitivity Hoefler et al. (2021); Evci et al. (2020); Constantin et al. (2018).

In addition to comprehensive surveys previously discussed that have summarized the landscape of sparse training techniques, another analyzed structured, unstructured, and dynamic strategies, primarily in the domain of convolutional neural networks (CNN) He & Xiao (2023). Although these reviews mention a small set of "sparse-from-scratch" methods, their design principles diverge from our neuro-inspired formulation in both computational overhead and energy considerations. For example, GraSP Wang et al. (2020) evaluates pruning candidates via second-order gradient information, an approach that is computationally demanding. In contrast, NIST establishes a sparse training

protocol directly from biologically grounded principles without requiring such costly calculations, which usually lead to longer training.

Other approaches focus on architectural components, such as CATRO, which identifies channel importance through class-aware trace ratio optimization Hu et al. (2023). Although effective in compressing CNNs and accelerating inference, CATRO is tailored toward convolutional layers and inference efficiency, whereas NIST targets multilayer perceptrons and emphasizes energy efficiency throughout the learning process.

In parallel, gradual and dynamic pruning remain central themes in the literature. Fontana et al. Fontana et al. (2024) integrated progressive pruning with knowledge distillation to sustain predictive accuracy, and Wang et al. Wang et al. (2023) developed a relaxation-based, layer-wise pruning strategy to continuously enforce sparsity.

Moreover, recent surveys have emphasized a growing line of research that seeks inspiration from biological systems in order to guide the development of sparse learning algorithms Jiao et al. (2022). This perspective has underscored a number of open challenges, such as the tendency of pruning methods to eliminate discriminative or edge-related features, and the broader difficulty of integrating sparse representations with modern deep architectures. These challenges highlight a tension between biological plausibility and engineering practicality: while sparsity is essential for efficiency, naïve pruning can undermine the representational capacity of the network.

Beyond pruning, new design paradigms for artificial neural networks have been suggested, particularly those that integrate heterogeneous modules or leverage architectural diversity to improve efficiency and robustness Shao & Shen (2023). Such ideas point toward the possibility of moving away from monolithic dense models toward architectures that incorporate structural bias in more principled ways.

Motivated by these insights, our work proposes a bio-inspired sparse-from-scratch algorithm that avoids the pitfalls of feature elimination entirely. Instead of discarding inputs or aggressively pruning connections, our approach maintains full feature coverage while enforcing sparsity through topographical connectivity constraints. By doing so, it provides a biologically grounded path toward efficient deep learning that reduces training cost while preserving accuracy and representational richness.

Feature-dimensionality reduction has been widely investigated as a route to improve neural network sparsity, often by selecting or discarding input features (feature pruning). Such methods typically require explicit evaluation of feature importance, for example, by statistical measures, gradients, or through auxiliary optimization routines, which introduces overhead in both computation and design complexity Hoefler et al. (2021); Zhou et al. (2021); Rao et al. (2023). These approaches may reduce inference cost, but they risk losing possibly useful information early, and their extra feature-selection steps may offset the gains in efficiency.

In contrast, in our NIST framework, we preserve all input features. Instead of discarding any input neurons, we introduce a 'topographical sparse layer', setting the neuroseed factor of the first layer at 1, which enforces sparsity at the connection level, matching the number of synaptic links with the number of input features. This ensures that every feature remains represented, while still achieving reduced connectivity and lower computational burden—without requiring separate feature importance ranking or pruning heuristics.

The existing literature suggests a clear need for sparse learning algorithms that are biologically plausible, preserve representational richness, and lower training cost. Many current sparse / pruning techniques depend on computationally expensive criteria (gradient-based, magnitude, sensitivity, etc.) to decide which weights, features, or neurons to remove. These can introduce instabilities or require hyperparameter tuning, especially when pruning early in training.

NIST addresses these weaknesses by adopting a biologically inspired connectivity scheme rooted in a form of retinal topography. Rather than evaluating neurons or connections for significance using loss-based or gradient-based metrics, NIST fixes a sparse connectivity pattern from the start, enforcing connectivity constraints that mimic biological architecture. This biases the learning process to distribute capacity more evenly, reduces the need for iterative pruning, and yields competitive performance even at high sparsity, while keeping training computation and energy cost significantly lower than methods that depend on dynamic or feature-pruning routines.

An especially promising avenue for future exploration is the integration of NIST with token sparsification techniques in transformer architectures. While NIST enforces biologically inspired connectivity sparsity at the synaptic level, token sparsification operates at the sequence level by dynamically reducing the number of tokens processed during self-attention. The combination of these complementary strategies has the potential to achieve significantly higher overall sparsity and compression, reducing both memory footprint and computational demand. Such a hybrid framework could pave the way toward highly efficient large-scale models that retain strong performance while operating under stringent energy and resource constraints.

## A.2 METHODOLOGICAL ASPECTS

Although multilayer perceptrons (MLPs) are highly simplified compared to biological neural circuits, it is still possible to establish meaningful parallels. In our analogy, the input layer of an MLP is treated as a *topographical map*, reminiscent of the way sensory systems such as the retina project information to downstream processing areas in the brain. Each input neuron corresponds to a specific feature or sensory element, and its connectivity to the first hidden layer represents the initial stage of synaptic projection.

In conventional MLPs, layers are fully dense: every neuron connects to all units in the preceding and subsequent layers. This is unlike the brain, where connectivity is often sparse and structured. In particular, retinal projections to the visual cortex exhibit *retinotopy*: local patches of the visual field are mapped to local patches in cortex through sparse, one-to-one or convergent connections. Inspired by this, we propose an input mapping where each input feature forms exactly one synapse to the hidden layer ($k = 1$). This design guarantees that no feature is discarded while minimizing density and parameter count.

Fig. A1 illustrates this biological analogy. Each portion of the input space projects to subsequent processing units via a minimal connection pattern, while the hidden layers retain denser connectivity. The resulting architecture mirrors two biological principles:

1. **Feature preservation:** every input is guaranteed to project into the network, avoiding feature loss at the first stage.

2. **Parameter efficiency:** by constraining the first layer to be sparse, the total parameter count is greatly reduced, removing the need for additional optimization of indexing, initialization, or pooling strategies.

This mapping is particularly relevant for architectures with stacked MLP layers, such as classifier heads in CNNs (e.g., VGG16, AlexNet). In these cases, setting the first fully connected layer to $k = 1$ effectively replaces global average pooling or max pooling, while still ensuring complete coverage of all extracted features. Subsequent layers may remain denser, consistent with the observation that certain cortical regions (e.g., the neocortex) exhibit higher synaptic densities than early sensory projections.

In summary, the proposed *topographical sparse input mapping* is motivated by the convergent organization of the visual system: sparse, one-to-one projections at the sensory interface, followed by denser integration in higher cortical stages. Translating this principle into ANN design enables substantial parameter savings at the input stage while preserving performance and biological plausibility.

## A.3 TRAINING DETAILS

Table A1 summarizes the training configurations used in our experiments. For each architecture, we report the optimizer, number of training epochs, batch size, and learning rate schedule. We adopted standard settings commonly used for each architecture (e.g., 80% of epochs with Adam and 20% with SGD+momentum for CNNs and AdamW+CosineLRScheduler for Transformers), with learning rate schedules tailored to the total number of epochs. This table provides a consolidated overview of the hyperparameters underlying all results presented in the main text.

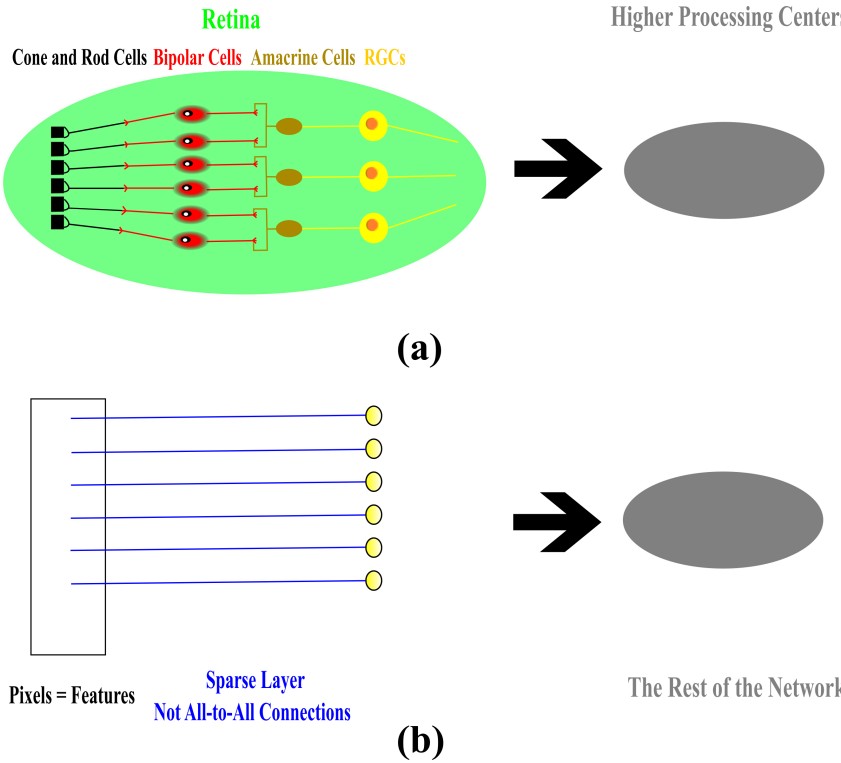

**(a)**

**(b)**

Figure A1: Topographical input mapping for stacked MLPs. (a) Simplified illustration of retinotopy, where different cell types are organized after the input layer. (b) Analogy in MLPs, where inputs are mapped with sparse, non-all-to-all connectivity. Following the retinal topography of the mammalian visual system, all input features are preserved but projected sparsely into denser subsequent layers.

Table A1: Training details for experiments.

| Model | Optimizer | Epochs | LR | Batch Size |
|-------|-----------|--------|-----|-----------|
| LeNet-5 | Adam + SGD(momentum=0.9) | 120 | 1e-3 | 64 |
| AlexNet | Adam + SGD(momentum=0.9) | 60 | 1e-3 | 64 |
| VGG16 | Adam | 60 | 1e-4 | 64 |
| MobileNetV1 | Adam | 30 | 1e-3 | 64 |
| DeiT_small | AdamW(wd=0.05) | 400 | CosineLRScheduler( initial_lr=1e-3, lr_min=1e-4, warmup_lr_init=lr / 25, warmup_t=warmup_epochs=5, cycle_decay=0.1, cycle_limit=1 | 128 |

## A.4 SUPPLEMENTARY MATERIALS

### A.4.1 NIST + LENET-5

To further evaluate the generality of our sparsification method, we tested it on LeNet-5, a classical CNN architecture (An et al., 2024). Although simple, LeNet-5 provides valuable insights into whether our approach is applicable to convolutional architectures with fully connected (FC) classifier heads.

**Architecture.** LeNet-5 processes $32 \times 32 \times 3$ color images and consists of three convolutional layers interleaved with subsampling (average pooling), followed by two FC layers (84 units and 10 units, respectively). The final convolutional stage produces 120 features, which are passed to the first FC layer. While the two FC layers account for $\sim$18% of the total parameters (11,014 out of 60,806), we observed that sparsifying this part of the network improved generalization.

**NIST applied to LeNet-5.** We applied NIST exclusively to the two FC layers, aiming to prune them aggressively while preserving classification accuracy. Four experimental configurations were compared: 1. The original dense model, 2. NIST with 98% sparsity, 3. NIST with 99.5% sparsity, 4. NIST with 99.5% sparsity initialized from a 95% sparse configuration.

Models were trained on both FMNIST and CIFAR-10 datasets for 20 runs each, and we report 95% confidence intervals and maximum validation accuracies (see Table A2).

Table A2: Results of applying NIST sparsification to the fully connected layers of LeNet-5 on FM-NIST and CIFAR-10. We report mean $\pm$ margin of error (95% CI) validation accuracy, maximum validation accuracy, final sparsity, and compression ratios over 20 runs.

| Model | Neuroseed Factors | Init. Sparsity (%) | Mean $\pm$ MOE Acc. | | Max Acc. (%) | | Final Sparsity (%) | Comp. Ratio |
|---|---|---|---|---|---|---|---|---|
| | | | FMNIST | CIFAR-10 | FMNIST | CIFAR-10 | | |
| LeNet-5 (Dense) | – | 0 | $91.3 \pm 0.001$ | $58.2 \pm 0.003$ | 92.0 | 60.1 | 0 | 0 |
| LeNet-5 + NIST | 1, 10 | 91.2 | $90.9 \pm 0.002$ | $58.7 \pm 0.050$ | 92.3 | 60.9 | 98 | 50 |
| LeNet-5 + NIST | 1, 10 | 91.2 | $91.2 \pm 0.002$ | $59.8 \pm 0.002$ | 92.1 | 61.8 | 99.5 | 200 |
| LeNet-5 + NIST | 1, 5 | 95.0 | $90.9 \pm 0.005$ | $59.4 \pm 0.042$ | 92.3 | 61.4 | 99.5 | 200 |

**Key findings.**

- **Accuracy improvements under extreme sparsity.** Remarkably, the 99.5% sparse model slightly *improved* mean validation accuracy on CIFAR-10 (58.2% $\rightarrow$ 59.8%), demonstrating that extreme sparsification can regularize training and mitigate overfitting.

- **High sparsity from scratch.** By setting neuroseed factors of 1 and 10 for the two FC layers, the model began training with 91.2% sparsity. Using a neuroseed factor of 5 for the output layer increased initialization sparsity to 95%. After only four epochs, a final pruning stage lifted sparsity to 99.5%.

- **Dynamic pruning stability.** NIST's single-shot pruning at epoch 4 stably increased sparsity to 99.5% without destabilizing training, confirming the robustness of the method under tight parameter budgets.

- **Reduced overfitting.** While validation accuracies on FMNIST remained similar across settings, training–validation curves revealed striking differences. Figure A3 shows that the dense baseline overfit substantially, as indicated by a large gap between training and validation accuracy. In contrast, NIST-pruned models exhibited narrower gaps, consistent with improved generalization.

Overall, these experiments demonstrate that NIST can prune LeNet-5's FC layers to extreme levels of sparsity (up to 99.5%) while preserving or even improving performance. This highlights both the computational efficiency and the regularization benefits of our framework, even on relatively small-scale CNNs.

For a more detailed illustration, Figs. A2 and A3 are provided, depicting learning dynamics and the quantified overfitting for the four model configurations described above. The red trace, corresponding to the original dense model, has the highest degree of overfitting, indicated by a significant gap between training and validation accuracy. This comparison is especially meaningful because the validation accuracies across the models are quite similar, with NIST being a bit higher. The dense model overfits by learning non-essential patterns present in the training data.

Although both models (original and NIST 99.5%) reach similar final validation accuracies, the original model, exhibiting a larger gap between training and validation curves, demonstrates signs of

overfitting. This gap indicates that the model fits the training data more tightly, capturing noise or spurious patterns that do not generalize well to unseen data (see Figs. A2 and A3).

In contrast, a NIST-sparsified model with a smaller training-validation gap tends to learn more generalizable features, resulting in better generalization behavior and greater robustness across different runs or datasets, even if the final validation accuracy is numerically similar.

Surprisingly, the highly sparsified model outperforms the dense baseline on the more challenging dataset, CIFAR-10. This suggests that NIST acts not only as a sparsifier but also as a regularizer, possibly helping avoid overfitting or improving generalization.

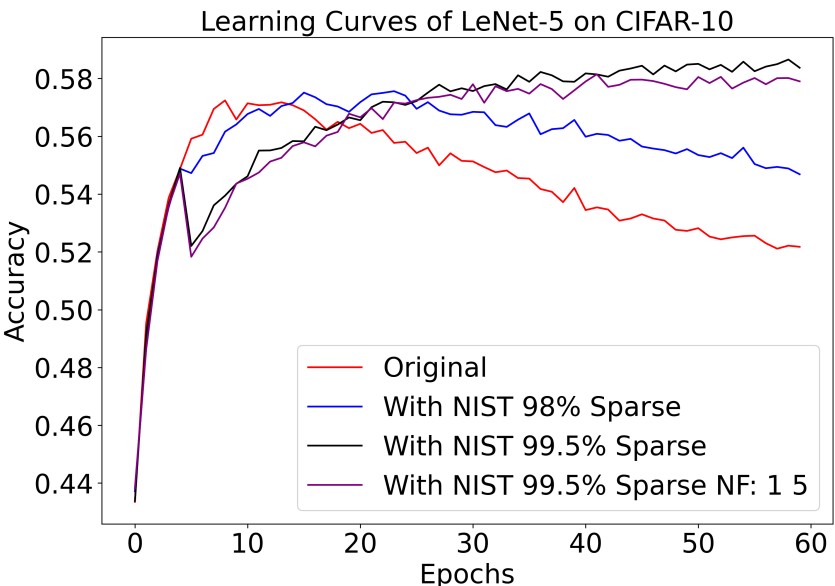

Figure A2: Validation curves on CIFAR-10 over 60 training epochs for four models with varying levels of NIST sparsification. The dense model (red) exhibits significant overfitting, while the 98% sparsified model (blue) shows improved stability. The two 99.5% NIST-pruned models demonstrate consistent performance and reach a plateau, indicating enhanced generalization and reduced overfitting through effective parameter allocation. The black curve exhibits a higher final validation accuracy since it starts with a greater neuroseed factor for the second layer (10 instead of 5).

### A.4.2 NIST + ALEXNET

Deep convolutional networks owe much of their success to overparameterized fully connected (FC) layers, yet these layers contribute disproportionately to model size and memory footprint. While traditional pruning methods seek to compress networks post hoc, they offer little guidance on how to reallocate that freed capacity for greater efficiency or accuracy. Above 90% of the parameters in some CNN architectures, such as AlexNet Krizhevsky et al. (2012), belong to FC layers, which serve as the last classifying layers. Here, we apply NIST on AlexNet to investigate how its performance varies when sparsified by NIST.

**Experimental setup.** We considered comparing five simulation scenarios for AlexNet in classifying the CIFAR-10:

- Dense baseline
- 98% sparsity via NIST with neuroseed factors 1, 2048, 10 (50% initial sparsity)
- 99.5% sparsity via NIST with neuroseed factors 1, 2048, 10 (50% initial sparsity)
- 99.5% sparsity with neuroseed factors 1, 50, 10 (99.25% initial sparsity)
- 99.99% sparsity with neuroseed factors 1, 2, 10 (99.84% initial sparsity)

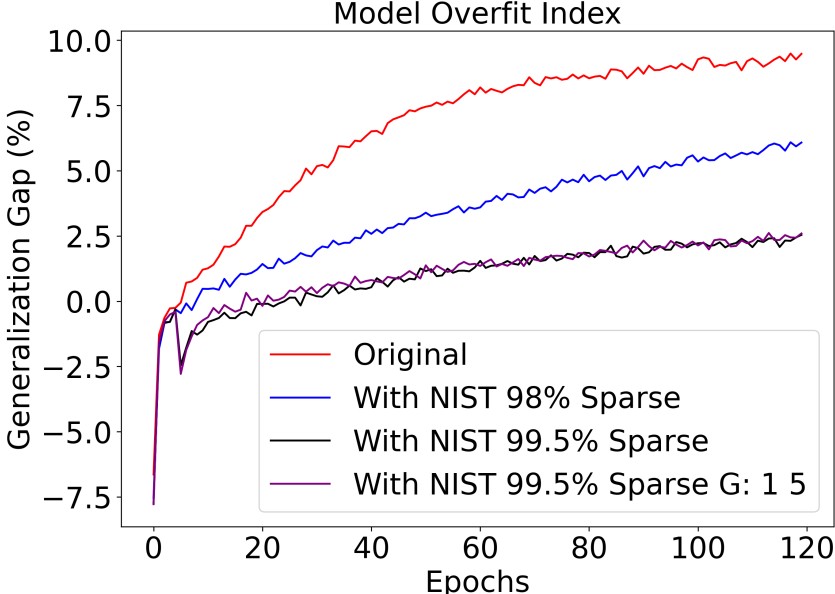

Figure A3: Evolution of the train–validation accuracy gap during training. The dense (original) model shows increasing overfitting (widening gap), while the NIST-pruned models exhibit a consistently smaller generalization gap, indicating improved regularization. The purple line represents a model similar to the black one, with a difference in neuroseed factors. For the purple, NF: 1 5 indicated neuroseed factors of 1 and 5 for the first and the second FC layers, respectively.

Please note that AlexNet in this experiment has a total of $\sim 35.8M$ parameters since we modified it for CIFAR-10 with 64, 192, 384, 256, and 256 filter sizes for the first to fifth convolutional layers with $3 \times 3$ kernels. We ran each of the five configurations over 20 independent simulations, tracking both training and validation accuracy curves. We report the mean and standard deviation of the peak validation accuracy for each scenario.

Table A3 contains the results of sparsifying AlexNet's FC layers with NIST. It has the four sparsification scenarios described in the previous section, with extreme sparsity ratios ranging from 98% to 99.99%. Since there are 3 FC layers in the classifier head of AlexNet, three corresponding neuroseed factors are assigned, one for each layer. By definition, the first layer, which receives the extracted features from the convolutional backbone, has a neuroseed factor of 1 in all cases.

Mean validation accuracies are calculated by averaging the peak validation accuracy from 20 independently trained models per scenario. The max accuracy is the maximum validation accuracy of each scenario across all simulated models. Compression ratios indicate the degree to which the size of a neural network's head is reduced through pruning. Initial sparsity ratios are achieved by the choice of neuroseed factors from scratch, while the final sparsity ratios indicate the sparsity after additional pruning during training. In the fifth case, there is no pruning, and 99.99% sparsity is achieved solely by neuroseed factors from the start.

Fig. A4 presents the learning dynamics of AlexNet and its NIST-sparsified modifications. In the two subfigures, the original dense model is compared against four sparse configurations. In Fig. A4A, the purple curve, representing 99.25% initial sparsity and 99.5% final sparsity, achieves the fastest convergence among all models. Overall, all sparsified models converge more quickly than the original. Despite differences in sparsity, the final training accuracies remain comparable across all cases. In general, reducing growth factors, which increases initial sparsity, leads to faster convergence in AlexNet.

Fig. A4B presents the corresponding validation accuracy curves. While the original dense model plateaus around 77%, all sparse variants exhibit superior generalization. The 99.99% sparse model performs best, attaining a validation accuracy of 85%. Unlike training accuracies, the validation

accuracies of NIST-sparsified models significantly surpass those of the original model, indicating greater generalization.

### A.4.3 THE NOVELTY AND CONTRIBUTION.

While AlexNet is no longer considered efficient by today's standards, the choice of architecture is not central to our objective. Our focus is on analyzing the sparsification dynamics of NIST, specifically as applied to fully connected (FC) layers. AlexNet serves as a valuable testbed due to its deep and overparameterized FC layers, offering a challenging and high-capacity environment for sparsification Fan et al. (2025).

Demonstrating that NIST can achieve extreme sparsity, up to 99.99%, while even improving accuracy in such a setting, provides strong evidence that NIST is robust and effective. These results suggest that NIST can generalize well to any architecture that incorporates FC layers, regardless of its overall design or modern relevance. Besides, AlexNet itself is still powerful and used in critical areas like medical image analysis Goyal et al. (2024); Medhat et al. (2024); Siuly et al. (2024); T. & R. (2024).

Also, this experiment shows that NIST is not just about pruning, but also about smart capacity optimization and generalization, which is even more compelling. So, for architectures that are highly overparameterized, like AlexNet, using NIST to sparsify from scratch yields consistent accuracy improvements and faster convergence (over baseline).

Our experiments demonstrate that not only can we prune up to 99.99% of classifier weights, but this extreme sparsification, indeed from scratch, ..., actually improves accuracy from $\approx 77\%$ to $\approx 85\%$, greatly surpassing the dense baseline.

These results validate our two-pronged thesis:

- Compression: NIST can prune virtually all redundant weights in MLP-heavy heads without loss of accuracy.
- Reallocation: Starting from an already sparse head and allowing controlled growth yields net performance gains, even under ultra tight parameter budgets.

Together, they position NIST as a versatile toolkit for both efficient deployment and architectures that demand maximum accuracy per parameter.

Table A3: Results of applying NIST sparsification to AlexNet's FC layers on CIFAR-10. We report mean $\pm$ margin of error (95% CI) validation accuracy, maximum validation accuracy, final sparsity, compression ratios, and parameter counts.

| Model | Neuroseed Factors | Init. Sparsity (%) | Mean Acc. $\pm$ MOE | Max Acc. (%) | Final Sparsity (%) | Comp. Ratio | Param Count |
|---|---|---|---|---|---|---|---|
| AlexNet (Dense) | – | 0 | 78.49±0.02 | 79.45 | 0 | 0 | 33,595,392 |
| AlexNet + NIST | 1, 4096, 10 | 50 | 82.98±0.01 | 83.35 | 98 | 50 | 671,907 |
| AlexNet + NIST | 1, 4096, 10 | 50 | 83.06±0.01 | 83.32 | 99.5 | 200 | 167,976 |
| AlexNet + NIST | 1, 50, 10 | 99.25 | 83.87±0.02 | 84.90 | 99.5 | 200 | 167,976 |
| AlexNet + NIST | 1, 2, 10 | 99.84 | 85.09±0.02 | 86.48 | 99.99 | 10,000 | 3,359 |

### A.4.4 NIST + VGG16

VGG16 is a deep convolutional neural network architecture proposed by the Visual Geometry Group at the University of Oxford, known for its simplicity and effectiveness in image classification tasks

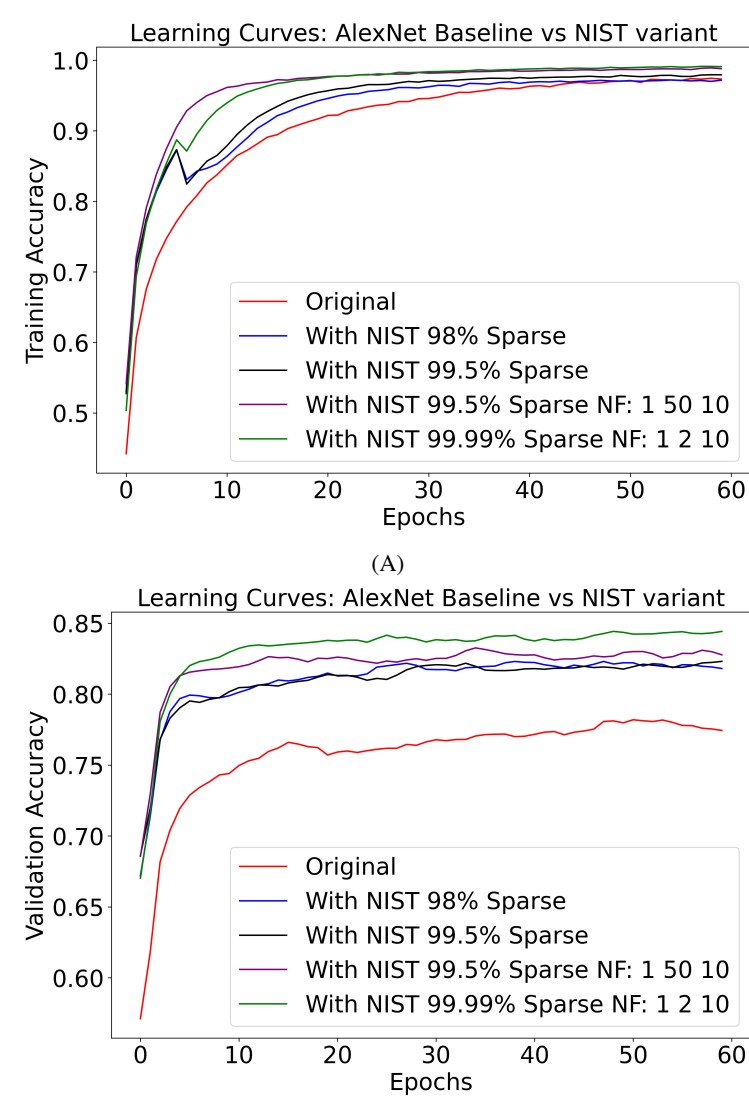

(A)

(B)

Figure A4: Learning curves of the original AlexNet and its sparsified modifications by the NIST algorithm. A) training accuracy vs epoch, and B) validation accuracy vs epoch. Validation accuracy curves for the sparsified models significantly surpass that of the original model (red curve). By reducing the neuroseed factors of the three fully connected layers to 1, 2, and 10, an initial sparsity of 99.84% was achieved from scratch. Continued pruning during training further increased sparsity to 99.99% (green curve). This extreme level of sparsification results in the highest validation accuracy and improved generalization, demonstrating effective capacity optimization.

Simonyan & Zisserman (2015). It consists of 16 weight layers, including 13 convolutional layers and 3 fully connected layers, with small 3×3 convolution filters and 2×2 max-pooling layers applied consistently throughout the network.

This design emphasizes depth while maintaining manageable computational complexity. VGG16 achieved state-of-the-art performance on the ImageNet dataset, demonstrating that increasing network depth with small filters can significantly improve classification accuracy. Its modular and uniform architecture has made it a popular baseline for transfer learning and a foundational model in various computer vision applications Tripathi et al. (2024).

About 90% of the parameters in VGG-16 belong to its FC layers, which serve as the last classifying layers. Here, we apply NIST on VGG-16 to investigate how its performance varies when sparsified by NIST. For our experiments, we use ImageNet100, a large-scale, real-world image classification dataset. ImageNet100 is a curated subset of the full ImageNet-1k dataset, consisting of 100 randomly selected classes. Each class contains approximately 1,300 training images and 50 validation images, maintaining the diversity and complexity of the original dataset while reducing computational requirements. We obtained this version of the dataset from Kaggle.

Each image in the dataset is a 3-channel RGB image with a resolution of $224 \times 224$ pixels. Preprocessing follows VGG's input pipeline in TensorFlow, which first converts images from RGB to BGR format and then zero-centers each color channel based on the ImageNet dataset statistics, without applying any scaling.

To train VGG-16 from scratch on ImageNet100, we freeze all convolutional layers except for the final convolutional block (block 5). The frozen layers retain their pretrained weights from ImageNet-1k, while block 5 and the classifier head are trained to adapt the model specifically to the ImageNet100 dataset. Given that the base model is pretrained, convergence was relatively fast, typically achieved within 30 epochs.

We observed that dense models with a large number of trainable parameters required lower learning rates to ensure stable training. In contrast, sparse or scratch-trained models benefited from higher learning rates, allowing for more aggressive updates to the limited set of trainable weights.

**Device.** We trained our models on an Nvidia RTX 3090 24GB GPU using TensorFlow.

**Sparsification scenarios.** We considered comparing three simulation scenarios:

- Dense baseline

- 99.63% sparsity with neuroseed factors 1, 2, 100 (Fully sparse from scratch)

- 99.9% sparsity with growth factors 1, 2, 100 (99.63% initial sparsity

**Results** Table A4 contains the results of sparsifying VGG16's FC layers with NIST as well as its original baseline. Since there are three FC layers in the classifier head of VGG16, three corresponding growth factors are assigned, one for each layer. By definition, the first layer, which receives the extracted features from the convolutional backbone, has a growth factor of 1 in all sparse cases.

As always, mean validation accuracies are the average of several independent trained models for each case. The max accuracy is the maximum validation accuracy of each scenario across all simulated models. Compression ratios indicate the degree to which the size of a neural network's head is reduced through pruning. Initial sparsity ratios are achieved by the choice of growth factors from scratch, while the final sparsity ratios indicate the sparsity after additional pruning during training.

Table A4: Performance analysis of NIST on VGG16 across two extreme sparsification scenarios, both above 99.6% sparsity from scratch. Using growth factors of 1, 2, and 100 in the final fully connected layers, the model achieves 99.63% sparsity from scratch. Pruning 99.9% of the trainable parameters led to the highest accuracy among all three cases.

| Model | Neuroseed Factors | Init. Sparsity (%) | Mean Acc. ± MOE | Max Acc. (%) | Final Sparsity (%) | Comp. Ratio | Param Count |
|---|---|---|---|---|---|---|---|
| VGG16 (Dense) | – | 0 | 71.2 ± 0.04 | 79.45 | 0 | 0 | 119,955,556 |
| VGG16 + NIST | 1, 2, 100 | 99.63 | 78.1 ± 0.02 | 78.34 | 99.63 | 270 | 443,835 |
| VGG16 + NIST | 1, 2, 100 | 99.63 | 78.2 ± 0.01 | 78.57 | 99.9 | 1000 | 119,995 |

**Computational efficiency.** Since NIST does not include any calculations or optimization techniques for pruning weights or regrowing them during training, there is no computational overhead and the total FLOPs for an MLP classifier head would be like Eq. 3:

$$\textbf{Training FLOPs} = 4 \times \text{active weights} \times \text{training samples} \times \text{epochs} \qquad (3)$$

This accounts for:

- 1× FLOP for the forward pass,
- 2× FLOPs for the backward pass (gradients),
- 1× FLOP for the weight update (e.g., optimizer step),
- → Total: 4× per active weight per sample per epoch

So, the computational cost analysis is provided in Table A5.

Table A5: Comparison of computational efficiency for different VGG16 head configurations. NIST achieves substantial reduction in FLOPs and active weights compared to the dense baseline.

| Configuration | Active Weights | Epochs | Total FLOPs $(\times 10^{12})$ | Rel. to Dense (%) | Notes |
|---|---|---|---|---|---|
| Dense VGG16 head (FC 25088→4096 →4096→100) | 119,995,556 | 20 | 959.9 | 100 | Baseline |
| NIST: sparsified from scratch | 443,835 | 5 | 0.88 | 0.091 | Reaches peak accuracy by epoch 2–3 |
| NIST: sparse + pruning after 1 epoch | 443,835 → 120,000 | 5 | 0.31 | 0.032 | Most efficient case |

Table A5 highlights the computational benefits of applying NIST to the VGG16 classifier head compared to the dense baseline. The dense configuration, with nearly 120 million active weights, requires about $9.6 \times 10^{14}$ FLOPs over 20 training epochs, serving as the reference point. In contrast, the NIST-based sparsified model reduces the active weights by more than two orders of magnitude, lowering the computation to just $8.8 \times 10^{11}$ FLOPs, while still reaching peak accuracy within only 3–4 epochs.

The most efficient case is achieved when pruning is applied after one epoch, reducing the weights further to 120k and cutting the total FLOPs to $3.1 \times 10^{11}$ which is only 0.032% of the dense baseline. These results emphasize that NIST not only accelerates training but also achieves drastic reductions in computational cost while preserving model. This makes VGG16's head cheaper than even tiny MLP heads in efficient networks, without sacrificing accuracy, a true revival of an otherwise "dead" architecture.

When sparsified with NIST, the model converges faster that usual. The combination of ultra-high sparsity and early convergence allows NIST to dramatically outperform the dense baseline in terms of both performance and sustainability, offering a practical and biologically inspired solution to reducing the carbon footprint of deep learning.

**NIST uncovers the "critical subnetwork"** By pruning away $\sim 99.9\%$ of the weights of the classifier head, which accounts for $\sim 88\%$ of the entire parameter count of the network, without any gradient-based retraining, we discover a leaner subnetwork that even improves performance. Our results echo the Lottery Ticket Hypothesis: large networks contain smaller, well-initialized subnetworks that can learn just as well when isolated Liu et al. (2024a); Malach et al. (2020).

**Low engineering barrier.** NIST teams with legacy VGG-based pipelines (e.g., in medical imaging, robotics, industrial vision) can adopt sparsification with minimal code changes, no need to re-architect or re-tune complex hyperparameters Veni & Manjula (2023).

## A.5 STATISTICAL ANALYSIS OF TRAINING SPEED & STABILITY (TEST ON MOBILENETV1)

In this section, we explored the application of NIST to already efficient architectures that use sophisticated mechanisms such as residual connections to reduce the parameter count. Models such as MobileNet and ResNet Howard et al. (2017); Shafiq & Gu (2022) already have fewer parameters compared to FC-heavy models like VGG. For these models, we tested the idea of expanding the classifier head (the last block of the architecture), while sparse-from-scratch, to keep the total parameter count the same or slightly lower.

When some models (e.g., MobileNet and ResNet) have tiny MLP heads, they have already reduced the parameter count by utilizing global average pooling to decrease the size of features fed to the classifier head. Although convolutional backbones compute most of the representational work, the classifier head plays a crucial role in mapping the backbone features to task labels. We are going to explore whether reorganizing the classifier head can improve optimization dynamics and generalization, specifically, by replacing a single fully-connected layer with several sequential NIST layers that together contain the same number of parameters.

For our experiments, we employed MobileNetV1, a lightweight convolutional neural network architecture designed for efficient inference on resource-constrained devices. MobileNetV1 is built upon depthwise separable convolutions, which factorize a standard convolution into a depthwise convolution followed by a pointwise convolution. This design significantly reduces the number of parameters and computational cost compared to traditional CNNs, while still maintaining competitive accuracy on image classification tasks. Due to its balance of efficiency and performance, MobileNetV1 serves as a strong baseline model for evaluating the effectiveness of our proposed classifier head modifications.

**Method settings.** We trained MobileNetV1 from scratch using the Adam optimizer with a learning rate of 0.001. In the original setup for CIFAR-10, the classifier head is a linear projection from the flattened feature vector of size 1024 to 10 output classes, resulting in $1,024 \times 10 = 10,240$ weights.

Instead of using an nn.Linear(10) layer, we replaced the head with a three-layer MLP of sizes 100, 200, and 10. The corresponding growth factors are 2, 50, and 10. This configuration yields 2,048 parameters for the first projection, 5,000 for the second, and 2,000 for the final projection, giving a total of 9,048 parameters, about 1,000 fewer parameters than the original dense head. So, in summary:

**Original configuration:** Convolutional base $\to$ Flatten layer $\to$ ($1 \times 1024 \to$ FC head (size=10)

**NIST alternative configuration:** Convolutional base $\to$ Flatten layer $\to$ ($1 \times 1024$) $\to$ NIST_layer(100, NF=2) $\to$ NIST_layer(200, NF=50) $\to$ NIST_layer(10, NF=10)

Here, NF denotes the growth factor. For example, the first NIST layer with a growth factor of 2 assigns two connections to each input feature, resulting in $1,024 \times 2 = 2,048$ weights, as noted earlier. Our choice of growth factors was designed to maintain roughly the same number of parameters as the dense head, while being sparser in the initial layers and more densely connected in the deeper layers, a pattern inspired by biological systems. We repeated each experiment for 20 independent seeds and measured: per-epoch validation accuracy, mean and standard deviation learning curves, and area-under-curve (AUC) of the validation accuracy vs epoch curve.

Across runs, the multi-layer sparse head consistently produced: (1) faster initial improvement, (2) $\sim 37.26\%$ reduction in standard deviation of validation accuracy, and (4) $> 37\%$ relative reduction in AUC variation (accuracy vs epochs).

## A.6 TRAINING DYNAMICS OF NIST + DEIT

Fig. A6 compares mean validation accuracy across 50 training epochs for the dense DeiT_small baseline and several sparse variants trained with NIST (50% sparse from scratch with different final sparsities). Two qualitative patterns stand out. First, all sparse counterparts climb noticeably faster during the initial phase of training (epochs 0– $\approx 30$), delivering substantially better validation accuracy than the dense baseline well before the mid-point of training. For instance, at epoch 10, the original baseline reached 26.3% accuracy, compared to $> 34\%$ for the sparse variants.

Table A6: Comparison of variability between Sparse-3 (NIST) and Dense-1 models across training epochs. Lower standard deviation (SD) indicates more stable training.

| Metric | Sparse-3 (NIST) Model | Dense-1 Model | Notes |
|---|---|---|---|
| Mean epochwise SD | 0.01627 | 0.02593 | Diff $= -0.00966$ ($-37.3\%$); Wilcoxon $p = 4.76 \times 10^{-4}$ |
| % epochs with lower SD (Wilson 95% CI) | 80% (0.627–0.905) | – | Proportion of epochs where $SD_A < SD_B$ |
| AUC_SD | 0.4880 | 0.7778 | Total variability across epochs: 37.2% |

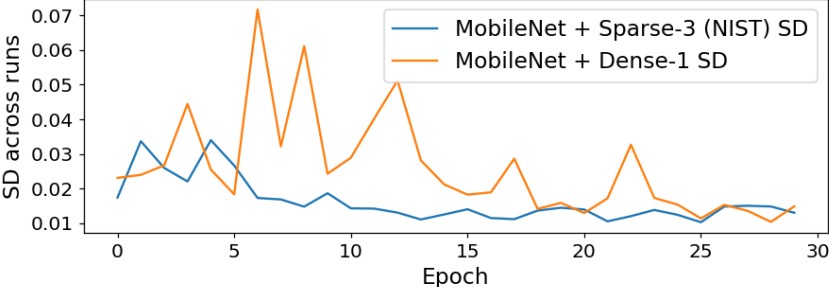

Figure A5: Standard deviation of validation accuracy across epochs for 20 independent runs of Sparse-3 Model (blue) and Dense-1 Model (orange). Each point shows the SD computed across the 20 runs at that epoch. Sparse-3 model consistently exhibits lower SD, indicating more stable training than Dense-1 Model.

Second, the dense model accelerates through the middle epochs and ultimately narrows the gap (and in some cases overtakes) by the end of training; nevertheless, the final accuracy of the dense model and the sparse variants up to $\approx 70\%$ sparsity are very similar (see Fig. 3B).

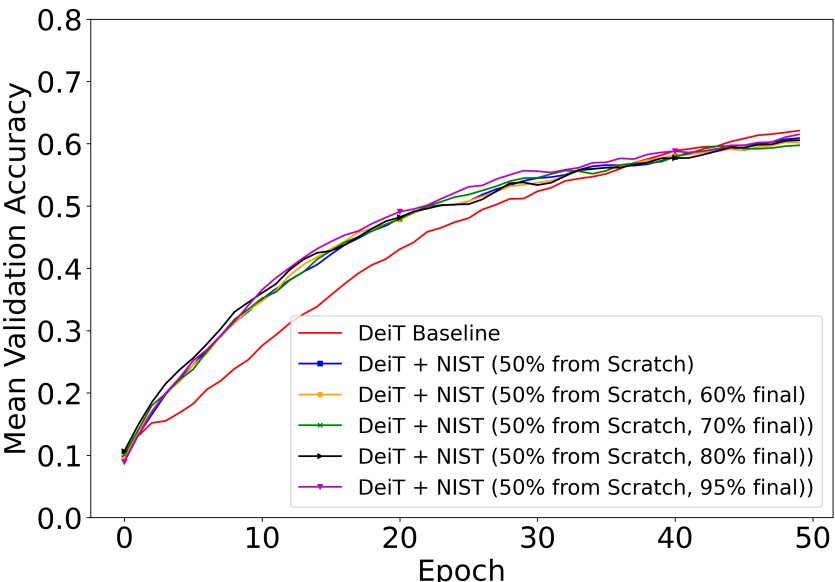

Figure A6: Learning dynamics of NIST on DeiT_small. Mean validation accuracy in the first 30 epochs is higher for sparse variants trained by NIST, indicating early training speed-up.

These results indicate that the principal practical benefit of the sparse models is their early training speed-up: when one cares about reaching a reasonable validation score quickly (for early stopping, model selection, prototyping, or compute-constrained scenarios), the sparse variants offer a clear advantage because they require fewer epochs (and typically less compute) to attain the same intermediate performance. In contrast, if the sole objective is the absolute best final convergence after long training, the dense model can recover and match or slightly exceed performance in later epochs. The differing dynamics — rapid initial gains for sparse models versus faster middle-epoch convergence for the dense model — suggest complementary trade-offs that can be exploited through hybrid schedules, early-stopping rules, or adaptive sparsity strategies depending on the application.

