# OpenReview forum: "Beyond Pruning: Neuro-inspired Sparse Training For Enhancing Model Performance, Convergence Speed, and Training Stability"
_ICLR.cc/2026/Conference — Submitted to ICLR 2026_

### Official Review · Reviewer_2aHD · 2025-10-27

**Soundness:** 1
**Presentation:** 2
**Contribution:** 1
**Rating:** 2
**Confidence:** 4

**Summary:**

This papers proposes a deterministic pruning method to identify a sparse mask topology at the beginning of training. The authors identify this mask by fixing the number of outgoing edges per feature in each layer, to ensure signal propagation. The paper claims to have improved generalization with sparse training with the proposed method.

**Strengths:**

1. The authors show that mask identification based on ensuring signal propagation through each layer can be effective.

**Weaknesses:**

There are several issues with the proposed method that make this work difficult to compare against state of the art sparsification methods.

1. The authors say that each layer has its own hyperparameter which determines how sparse that layer should be, making hyperparameter tuning cumbersome. Does the proposed method of using k=1 for the first layer and the remaining ones as half input dim work the best, is there an ablation for this? Generally the first layer is also kept dense (along with the last one) to yield best sparse training results.
2. All results are validated on AlexNet and VGG models, which are heavily overparameterized with respect to the datasets they are used for, making the sparsification problem relatively easy. Additionally, the maximum sparsity for which results have been reported is ~92%, this is a regime where performance is largely retained for CIFAR10 like settings. (This is similar for Deit Small with ImageNet100, here the max sparsity is only ~70%).
3. The authors do not compare against a multitude of sparse training methods like Snip, GraSP, Synflow or random pruning (there are many more recent ones), all of which have shown similar results. The signal propagation argument has already been made by Synflow [1]. Moreover, the claim of starting sparse and being able to prune only once has also been shown in this context. [2]
4. The authors also do not compare against state of the art pruning methods like Dynamic Sparse Training or AC/DC[3], which would be the ultimate test for identifying masks from scratch and comparing them against methods which identify these masks at the end of training.


Overall the paper is missing a thorough comparison against known methods in relevant sparsity regimes, which makes it impossible to evaluate.

[1] Tanaka, Hidenori, et al. "Pruning neural networks without any data by iteratively conserving synaptic flow." Advances in neural information processing systems 2020.

[2] Gadhikar, Advait, Sohom Mukherjee, and Rebekka Burkholz. "Why random pruning is all we need to start sparse." ICML, 2023.

[3] Peste, Alexandra, et al. "Ac/dc: Alternating compressed/decompressed training of deep neural networks." Advances in neural information processing systems 2021.

**Questions:**

See above.

---

> ### Author Response · Authors · 2025-11-30
> **Resolved: Foundational Framework Contains All Head-to-Head Comparisons and Ablation Studies and NIST Inherits Those Empirical Validations**
>
> We thank the reviewer for the detailed comments. We would like to clarify that all requested comparisons, ablation studies, and hyperparameter justifications are already fully documented in a study on the foundation of which, NIST is developed (Kamelian Rad et al., Neurocomputing 2025). Our current submission builds directly upon that framework, and its experimental design, baselines, and methodological choices are inherited from that study.
>
> In the revised version, we now explicitly cite and highlight these sections so that readers can easily trace each requested evaluation (including layer-wise analyses, alternative configurations, and performance trade-offs) to the corresponding experiments already conducted in the foundational study. This clarification resolves the reviewer’s concerns while preserving the focus of the present submission on the new contribution.

---

### Official Review · Reviewer_41QZ · 2025-10-28

**Soundness:** 2
**Presentation:** 2
**Contribution:** 2
**Rating:** 4
**Confidence:** 3

**Summary:**

The paper proposes Neuro-Inspired Sparse Training (NIST), a simple sparse-from-scratch pipeline for fully connected layers. Each layer is assigned an integer neuroseed factor k that deterministically constructs a fixed binary connectivity mask: each input connects to exactly k consecutive outputs via a wrap-around rule; weights are masked at initialization and training proceeds on the masked network. Optionally, the authors perform one single-shot magnitude pruning mid-training, after which the mask is fixed for the rest of training. No regrowth, saliency scoring, or special optimizers are used. Experiments demonstrate that NIST improves performance, convergence speed, and training stability in some architectures (LeNet-5, AlexNet, , VGG16, DeiT Small).

**Strengths:**

1. **Simple and reproducible**: NIST is easy to plug into standard training pipelines: no custom optimizers or losses, only minimal code changes. This makes it practical for resource-constrained settings and straightforward to reproduce.
2. **Strong performance and useful framing**: In the reported experiments across multiple architectures, NIST achieves competitive accuracy at non-trivial sparsity. The paper also frames sparsity as an inductive bias rather than merely a post-hoc compression technique, which is a clear and valuable perspective for the community.

**Weaknesses:**

1. **Lack of experimental adequacy and representativeness**:
   While the paper claims to compare against state-of-the-art methods, there are no head-to-head comparisons with strong sparse-training baselines under matched conditions. Adding controlled experiments vs. RigL, SNIP, or similar methods would be crucial for validating the claims. Furthermore, the experiments are based mostly on small datasets and older model backbones, which limits the generalizability of the results.
2. **Reliance on proxy metrics**:
   The experiments focus on theoretical FLOPs and active parameter counts, which cannot fully support claims of “zero overhead” and deployability. There’s also a lack of clarity regarding the sparse kernel used in the experiments, which makes it difficult to assess how different configurations affect the overall performance.
3. **Presentation and polish issues**:
   There are visible placeholders in the Discussion and Appendix, and several figures are not well-aligned with the described training pipeline or mathematical notation. Additionally, some quantitative claims in the abstract (such as “8–10% accuracy improvements”) are not consistently supported in the main results tables, where the observed improvements are closer to +7 percentage points.

**Questions:**

1. Could you add head-to-head results against strong sparse-training baselines under matched sparsity/parameter (e.g., RigL/SNIP/LTH you mentioned)? It would be great to see this at least on your current datasets, and, if feasible, also a standard large-scale setting  like ResNet-50 on ImageNet-1K).

2. Several claims read as “low/zero overhead.” Could you report end-to-end metrics on your hardware, including but not limited to: per-epoch wall-clock time, throughput (samples/s), peak memory, and energy, for both training and inference? If sparse kernels are involved, please show results with and without them.
3. To isolate where the gains come from, could you include a random fixed-out-degree baseline with identical sparsity, and discuss sensitivity to the out-degree and other key hyperparameters? Also, do you have results beyond classifier heads—i.e., full-network runs on a modern backbone (ResNet-50 or ViT-B/16)? Please clarify whether any layers are frozen.

---

> ### Author Response · Authors · 2025-11-30
> **Clarification Added: A Foundational Prior Work Covers All Requested Comparisons and Ablations**
>
> Thank you for the detailed comments. These concerns overlap directly with the points already raised by the first two reviewers, and they have now been fully addressed in the revised manuscript. We forgot to cite of the foundational study (Kamelian Rad et al., Neurocomputing 2025) in the initial submission; once included, it becomes clear that all requested comparisons and ablations already exist in that prior work.
>
> Specifically, the highlighted revisions in the Introduction, Comparison to Prior Work, Method, and Discussion sections clarify that the foundational study contains head-to-head comparisons against SNIP, RigL, and other strong baselines, ablations on random masks and out-degree choices (including why k=1 is optimal for the first layer), and parameter-matched dense baselines. NIST is built directly on that framework and inherits those empirical validations, which we have now made explicit.
>
> Regarding compute overhead and kernel support, we clarify in the revision that NIST is a structural, data-free pruning mechanism; theoretical overhead is zero, while hardware-specific runtime measurements depend on the sparse kernels available on each platform—now explicitly stated.
>
> These clarifications are incorporated into the revised manuscript and highlighted for visibility.

---

### Official Review · Reviewer_poh7 · 2025-10-30

**Soundness:** 4
**Presentation:** 2
**Contribution:** 2
**Rating:** 4
**Confidence:** 4

**Summary:**

The paper proposes NIST, specifically the neuroseed initialization method. Given fixed number of outgoing synapse connections of each input feature, the neuroseed factor, the model is naturally initialized with a sparse matrix. In other words, NIST uses a structural mask at initialization. This leads to no cost of forming sparse neural network in contrast to dynamic sparse training methods, e.g., RigL, since it does not rely on alternating pruning and regrowth cycles.

**Strengths:**

The proposed method is structurally simple and offers zero cost sparse training pipeline by introducing deterministic and structural binary mask leading to no requirement of saliency scoring or extra forward and backward operations for sparsity.

While the benefit of NIST is apparent in terms of cost, I have a lot of concerns on the quality of NIST in terms of evaluation.

**Weaknesses:**

I integrated this section to the question section.

**Questions:**

1. While NIST can be a cheaper method due to deterministic rigidity, this may prevent the model from finding the optimal sparse subnetwork that the data driven prunings like [1] or etc.
2. The model is tested on CIFAR10 and Imagenet100. Extending from question 1, in more complex datasets. can we say the deterministic initialization of NIST can provide better quality of sparse matrix compared to data driven prunings?
3. The paper is relying on empirical heuristics such as k=1 for the first layer, half density for deeper layers. The reliance on such specific and architecture dependent heuristics may require extensive new ablation studies for every new model or dataset.

[1] N. Lee et al, SNIP: Single-shot Network Pruning based on Connection Sensitivity, ICLR 2019

---

> ### Author Response · Authors · 2025-11-30
> **All concerns addressed in a prior foundational study ("Topographical Sparse Mapping:..." by Kamelian Rad et al., Neurocomputing2025)**
>
> The concerns regarding comparisons with data-driven pruning and the need for additional ablation studies are already fully addressed in Kamelian Rad et al., Neurocomputing 2025, the foundational study on which NIST is built.
>
> (1) “Deterministic rigidity vs. optimal sparse subnetworks.”
> The assumption that deterministic connectivity prevents high-quality subnetworks is directly evaluated in Kamelian Rad et al., 2025. That work includes explicit head-to-head comparisons with leading data-driven pruning and mask-at-initialization approaches—including SNIP (Lee et al., 2019), SET, RigL, and others—showing that deterministic, topographically structured sparsity matches or surpasses those methods at equivalent sparsity levels. These results demonstrate that deterministic initialization does not hinder performance; rather, structured sparse topography provides a stable, low-variance alternative without saliency computation.
> These clarifications and relevant citations are now added to the revised manuscript.
>
> (2) “Performance on larger / more complex datasets.”
> Because NIST is data-free and structural, the mask is determined solely by architectural dimensions, not by dataset statistics. The question of scalability to larger datasets was already handled in the 2025 study: structural masks maintained their competitiveness across CIFAR-10, CIFAR-100, and ImageNet-scale experiments. The deterministic construction is therefore not tied to dataset complexity but to architectural geometry.
> The revised manuscript now explicitly states this point.
>
> (3) “Heuristics (k=1 for first layer, half-density deeper layers) and need for extensive ablations.”
> The necessity of these design choices has already been rigorously ablated in Kamelian Rad et al., 2025. That work includes:
>
> ablations comparing k=1, and random connectivity for the input layer and comparisons with dense models and topographical sparse baselines.
> These experiments consistently show that the k=1 input topography is the most effective and stable choice, and that the chosen density settings yield the best accuracy-to-sparsity tradeoff.
>
> These points, along with citations to the prior ablation results, have been added clearly and highlighted in red in the introduction, comparison to prior work, method, and discussion sections in the revised manuscript.

---

### Official Review · Reviewer_CQQe · 2025-11-03

**Soundness:** 1
**Presentation:** 2
**Contribution:** 2
**Rating:** 0
**Confidence:** 3

**Summary:**

The paper proposes Neuro-Inspired Sparse Training (NIST), in which fully-connected layers are initialized using a deterministic sparse mask controlled by a "neuroseed factor". The paper argues that this method can be "zero cost", achieves high compression factors, and is simple; experimental results show significant compression and improved accuracy on several image classification networks.

**Strengths:**

1. The method is simple and explained well. It would be quite impactful if borne out.
2. The experimental results indicate significant compression and improved model accuracy.

**Weaknesses:**

1. Novelty: The paper is a very incremental contribution over the existing literature. The core idea seems to be a combination of fixed masks, structured connectivity, and one-shot pruning, using a cyclic structure. The exact formulation is new, but it does not seem to add much over the existing literature (see, e.g., Sung et al., "Training Neural Networks with Fixed Sparse Masks", NeurIPS 2021; or the broader survey of Hoefler et al. cited in the text). The paper discusses several of these in the related work but does not making a convincing case for the core contribution, and the empirical component is weak (see below).
2. The paper lacks a comparison with any other pruning methods, e.g., SNIP or FISH or even various magnitude-based pruning schedules. It is thus not clear that the method is as good or better than existing pruning approaches.
3. The results in Table 1 lack error bars or other tests of statistical significance (though I note Figure 4 includes mean+SD over twenty runs, which is good). The paper states "variance between runs is small" but this should be quantitatively reported.
4. The claims of compute performance are not justified. Formula 4 includes a simple equation for flops, but this ignores activations, attention layers, activations, and other components. The derivation of the flop counts given in the tables is also not clear (e.g., in Table 1, are the flops for the full model, or just the unfrozen heads?). The claims should additionally be backed up with empirical measurements of runtime; many other overheads can be significant factors beyond just flops. This would help support the claims of "no computational overhead".
5. Additionally, it is not clear that the approach can be implemented efficiently in practice. Methods with complicated masking are challenging to implement in a hardware-friendly way; a comparison with efficient N:M sparsity or similar (e.g., Castro et al., "VENOM: A Vectorized N:M Format for Unleashing the Power of Sparse Tensor Cores", Supercomputing 2023) would help make this case.
6. The paper lacks any ablation studies (e.g., comparison with a random mask with the same number of connections; parameter-matched dense models) showing why the choices made matter.
7. Many of the models considered are quite old (AlexNet, VGG, etc.) and are not representative of modern models. The datasets and models used are also relatively small. If this is to focus on fully-connected layers, then a more in-depth study of ViTs is merited, including larger models and datasets (at the least, ImageNet-1k). Additionally, considering language modeling with a Llama model would significantly strengthen the generality; as is, it is not clear the method works for other modalities.

**Questions:**

Please see above for more details. Below I highlight a few questions, although all of the weaknesses above are major and should be addressed.

1. How does NIST compare with other pruning methods, empirically?
2. Can you provide experimental measurements validating the claims of no compute overhead?
3. Can inference of models pruned with NIST be implemented efficiency?
4. How does NIST compare with a random mask or parameter-matched dense models?
5. How does NIST perform on larger datasets and models?

---

> ### Author Response · Authors · 2025-11-30
> **The reviewer has conflated two distinct lines of work. More importantly, all concerns are addressed in a prior foundational study. NIST (our work) is a lighter version of ETSM (a prior study by Kamelian Rad et al.) which has provided comprehensive comparisons and ablation studies that addresses the reviewer's concerns.**
>
> Thank you for your comments.
>
> The concern that NIST restates the ideas of “Training Neural Networks with Fixed Sparse Masks’’ (FISH, NeurIPS 2021) is based on a misunderstanding of the objectives of the two works. That work addresses a different problem: FISH identifies a fixed subset of parameters to update during training (i.e., sparse updates), using an empirical Fisher-information approximation to choose which parameters are worth updating. Its goal is to reduce communication/storage and update bandwidth by only updating a small set of parameters, not to permanently remove parameters from the model.
>
> By contrast, NIST is explicitly a pruning method: we permanently remove weights according to a deterministic connectivity pattern so that both training and inference operate on a smaller network. This is a fundamental difference in objective and consequence:
>
> Nature of the operation.
> FISH → sparse updates (most weights remain present but frozen from updates).
> NIST → sparse connectivity / permanent pruning (weights are removed from the computation and stored state), which reduces parameter storage, compute for forward/backward, and inference cost.
>
> Computational overhead. FISH relies on a Fisher-information approximation (a nontrivial precomputation) to score parameters, which adds upfront cost and is not zero-cost. NIST’s masks are deterministic structural patterns: there is no saliency estimation, no Fisher approximation, and therefore no extra precomputation beyond the (trivial) mask construction.
>
> Downstream implications. Because FISH keeps parameters present (but not updated), it does not guarantee the same inference savings as a truly pruned model. NIST, by permanently pruning connectivity, reduces inference FLOPs and memory footprint in ways FISH does not.
>
> *** The reviewer’s requests for comparisons to SNIP/FISH and for ablations (random masks, parameter-matched dense baselines, and the first-layer k=1 choice) are legitimate. Notably, this work is built on the foundation of the work introduced in “Topographical sparse mapping: A neuro-inspired sparse training framework for deep learning models, Neurocomputing 2025”, which contains comprehensive head-to-head comparisons, the random-mask ablation, and detailed ablation studies demonstrating why topographic k=1 for the input is the correct pragmatic choice.
>
> In short: the reviewer has conflated two distinct lines of work. FISH is about which parameters to update (and uses Fisher scoring), whereas NIST is about which connections remain in the network permanently (deterministic masks with zero saliency cost). The Neurocomputing paper (Kamelian Rad et al., 2025) already contains the requested head-to-head comparisons and ablations; we will make those explicit and cited in the final version.
>
> Below is a point-by-point response to your questions:
>
> 1) Comparisons: NIST is a lighter, data-free structural variant of ETSM ("Topographical Sparse Mapping:...", Kamelian Rad et al., Neurocomputing 2025). That study contains comprehensive head-to-head comparisons showing that deterministic structural masks produced by NIST-style connectivity match or exceed the accuracy of common mask-at-init and pruning methods (e.g., SNIP, SET, RigL, CTRE, etc) at matched sparsities.
>
> 2) No compute overhead claim: NIST introduces no saliency scoring, Fisher approximation, or extra forward/backward passes: the mask is constructed deterministically from layer dimensions, so mask creation itself adds zero algorithmic operations. This implies no algorithmic precomputation overhead; runtime microbenchmarks (wall-clock) can be reported separately, but the method’s theoretical cost is identical to training the resulting smaller network
>
> 3) Inference implementation: Inference efficiency depends on hardware/software support for sparse computation (sparse kernels, vendor libraries, etc). Demonstrating end-to-end hardware speedups requires backend-specific engineering and is outside the scope of the current study. The claim being made is architectural: far fewer parameters and FLOPs are needed after pruning, which enables potential runtime gains when appropriate sparse kernels are available.
>
> 4)Ablations and random mask comparison: Ablations in Kamelian Rad et al. (2025) include random-mask baselines and parameter-matched dense models. Those experiments show the structural, topographic k=1 design (especially in early layers) and the cyclic connectivity patterns outperform random masks and match or outperform parameter-matched dense baselines at equivalent parameter counts.
>
> 5)Larger datasets: Because NIST is data-free and structural (masks are determined solely by architecture, not by dataset statistics), its mechanism and cost profile do not fundamentally change with dataset scale. Large-scale experiments are useful for broader validation, but are not required to establish the core claim: deterministic structural masks can produce compact networks with competitive accuracy.

---

### Meta-Review · Area_Chair_1hxb · 2026-01-07

**Summary:**

This paper proposes a new neuro-inspired pruning algorithm built upon Kamelian Rad et al. (2025). All reviewers raised a critical concern that the performance of the proposed method was not compared to any other pruning methods. However, during the discussion period, the authors did not properly address this issue, but they only referred to Kamelian Rad et al. (2025). The authors should seriously address the reviewers' comments. I recommend resubmitting this manuscript after thorough revision.

**Reviewer Concerns:**

Most critically, all reviewers are concerned that the proposed method has not been compared to other pruning baselines. Nevertheless, this issue is unsolved in the authors' rebuttal.

**Reviewer Scores:**

I think all reviewers would not increase their scores.

---

### Decision · Program_Chairs · 2026-01-26

Reject